# Prediction of Historical, Current, and Future Configuration of Tibetan Medicinal Herb *Gymnadenia orchidis* Based on the Optimized MaxEnt in the Qinghai–Tibet Plateau

**DOI:** 10.3390/plants13050645

**Published:** 2024-02-26

**Authors:** Ming Li, Yi Zhang, Yongsheng Yang, Tongxin Wang, Chu Wu, Xiujuan Zhang

**Affiliations:** 1College of Horticulture & Gardening, Yangtze University, Jingzhou 434025, Chinawuchu08@yangtzeu.cn (C.W.); 2Key Laboratory of Adaptation and Evolution of Plateau Biota and Key Laboratory of Restoration Ecology in Cold Region of Qinghai Province, Northwest Institute of Plateau Biology, Chinese Academy of Sciences, Xining 810008, China; ysyang@nwipb.cas.cn

**Keywords:** medicinal plants, maxent model, driving factors, climate change, suitability

## Abstract

Climate change plays a pivotal role in shaping the shifting patterns of plant distribution, and gaining insights into how medicinal plants in the plateau region adapt to climate change will be instrumental in safeguarding the rich biodiversity of the highlands. *Gymnosia orchidis* Lindl. (*G. orchidis*) is a valuable Tibetan medicinal resource with significant medicinal, ecological, and economic value. However, the growth of *G. orchidis* is severely constrained by stringent natural conditions, leading to a drastic decline in its resources. Therefore, it is crucial to study the suitable habitat areas of *G. orchidis* to facilitate future artificial cultivation and maintain ecological balance. In this study, we investigated the suitable zones of *G. orchidis* based on 79 occurrence points in the Qinghai–Tibet Plateau (QTP) and 23 major environmental variables, including climate, topography, and soil type. We employed the Maximum Entropy model (MaxEnt) to simulate and predict the spatial distribution and configuration changes in *G. orchidis* during different time periods, including the last interglacial (LIG), the Last Glacial Maximum (LGM), the Mid-Holocene (MH), the present, and future scenarios (2041–2060 and 2061–2080) under three different climate scenarios (SSP126, SSP370, and SSP585). Our results indicated that annual precipitation (Bio12, 613–2466 mm) and mean temperature of the coldest quarter (Bio11, −5.8–8.5 °C) were the primary factors influencing the suitable habitat of *G. orchidis*, with a cumulative contribution of 78.5%. The precipitation and temperature during the driest season had the most significant overall impact. Under current climate conditions, the suitable areas of *G. orchidis* covered approximately 63.72 × 10^4^/km^2^, encompassing Yunnan, Gansu, Sichuan, and parts of Xizang provinces, with the highest suitability observed in the Hengduan, Yunlin, and Himalayan mountain regions. In the past, the suitable area of *G. orchidis* experienced significant changes during the Mid-Holocene, including variations in the total area and centroid migration direction. In future scenarios, the suitable habitat of *G. orchidis* is projected to expand significantly under SSP370 (30.33–46.19%), followed by SSP585 (1.41–22.3%), while contraction is expected under SSP126. Moreover, the centroids of suitable areas exhibited multidirectional movement, with the most extensive displacement observed under SSP585 (100.38 km^2^). This study provides a theoretical foundation for the conservation of biodiversity and endangered medicinal plants in the QTP.

## 1. Introduction

It is highly likely that global temperatures will exceed a 1.5 °C increase in the future [1]. The Intergovernmental Panel on Climate Change (IPCC) has emphasized in its Sixth Assessment Report (AR6) that surpassing the 1.5 °C threshold would cause the extinction of indigenous species and the loss of local habitats, posing a significant threat to future biodiversity [2]. High latitude regions, due to their cold climate, experience a faster warming rate than the global average. The effects of climate change on the plateau region are far-reaching, encompassing the melting of glaciers, a downward shift in snowlines, and the expansion of desertification, among others. These changes pose a significant threat to the region’s biodiversity [3]. High-altitude medicinal plants, which are integral to the plateau’s ecological makeup, are particularly vulnerable and face an elevated risk of extinction compared to others [4]. In order to further safeguard global biodiversity and foster regional economic development, it is imperative to comprehend the interplay between the potential geographic distribution of species and climate change. This knowledge will enable us to identify the potential geographic distribution areas for species in the face of future climate change scenarios and devise adaptive conservation strategies.

*Gymnosia orchidis* Lindl. (*G. orchidis*) is a vulnerable species (VU) belonging to the Gymnadenia genus in the *Orchidaceae* family. It holds the highest extinction risk among numerous plant species [5,6]. *G. orchidis* is known as “Wangla” in Tibetan medicine. The rhizomes of this plant contain compounds such as dactylorhin B and dactylorhin A [7]. It possesses significant medicinal value, including the treatment of lung deficiency, bleeding, and cough. It also plays a crucial role in the combined treatment of type 2 diabetes (T2DM), where its tuberous root and pumpkin seeds are utilized [8]. Relevant studies have indicated a positive correlation between the compounds found in *G. orchidis* and its growth age. Furthermore, moderate light intensity has been shown to enhance the production of chemical substances [7,9]. However, research on *G. orchidis* has primarily focused on its pharmacological actions, with limited publications on predicting its potential suitable zones [10]. This knowledge gap is concerning given the species’ longer growth cycle and reliance on wild resources, making it increasingly rare [8,11]. The wild resources of *G. orchidis* are continuously declining, with low production but increasing demand each year [12]. Therefore, urgent attention is required for the conservation and cultivation of *G. orchidis*. The temporal and spatial heterogeneity of *G. orchidis* is pronounced, with suitability continuously expanding from the MH (Mid-Holocene). The major distribution areas are situated in Qinghai, Yunnan, Tibet, and Sichuan, in China. With its growth environments being hillside forests and alpine grasslands at an altitude of 2800–4100 m, there is a wide variation in the quality of regional resources [11,13]. Hence, it is crucial to have a comprehensive understanding of the potential geographical distribution and the suitable habitat conditions of endangered medicinal plants as a prerequisite for conducting effective conservation work. However, in practice, there is a limited availability of data on the geographic distribution of endangered medicinal plants on the plateau region.

The Qinghai–Tibet Plateau (QTP), located at 67–105° E and 25–40° N, is a crucial region for *G. orchidis* distribution. With an average elevation of over 4000 m above sea level, the QTP exhibits distinctive characteristics, including high altitude, diverse vegetation distribution, and pronounced zonal differences [13,14,15]. The eastern and southeastern regions of the QTP are renowned for their abundant plant biodiversity. Nevertheless, the suitable habitat range for medicinal plants in these areas has been progressively shrinking, particularly in the eastern parts [16]. This decline can be attributed to the intricate topography and land characteristics of the region [17], as well as its plentiful water resources and favorable climate. Moreover, the relatively stable environment during glaciation periods has had an impact on the distribution patterns of medicinal plants [18]. These factors contribute to the creation of unique conditions that provide a suitable habitat for *G. orchidis*. Moreover, the QTP functions as a highly sensitive and vulnerable ecological screen to climate variability, not only in China but also across Asia as a whole [19]. Recent studies have indicated a warming and wetting trend in the QTP over the past 40 years, which is projected to continue. These changes not only impact the physiological and biochemical processes of various plant species, increasing the risk of extinction for endangered species like *G. orchidis*, but also have the potential to alter the global vegetation ecological pattern, posing significant threats to organism and ecosystem diversity [20]. Consequently, it is crucial to explore potential suitable areas for endangered species in the future to facilitate biodiversity protection and maintain ecosystem balance in this region [21].

The complex and diverse environment of the QTP presents difficulties in surveying numerous medicinal plants, and the absence of accurate species distribution maps adds to the challenges faced in conservation work [4]. Species distribution modeling (SDM) is employed as the primary method to simulate suitable habitat areas and predict distribution changes based on environmental variables [22]. Commonly used SDMs are the Genetic Algorithm for Rule-set Production (GARP), Climate and Expertise (Climex), and Maximum Entropy model (MaxEnt) [23,24]. Among the SDM approaches, the MaxEnt is widely used due to its simplicity, rapid processing, and high prediction accuracy, even with limited distribution points [25,26,27]. The MaxEnt has good applications in predicting prehistoric geology and future climate scenarios and is particularly suitable for predicting suitable areas for endangered species, medicinal plants, and alpine plants, as well as assessing their response to future climate changes [4,28,29,30].

This study focuses on *G. orchidis* and aims to simulate the current distribution pattern of its suitable areas in the QTP using the MaxEnt. Furthermore, it seeks to predict the potential suitable areas in paleo and future time periods to evaluate the spatial distribution of *G. orchidis* on a temporal scale. The ultimate goal is to develop a numerical model that accurately predicts the suitable areas for *G. orchidis*, facilitating informed introductions and resource conservation efforts. This study has the potential to catalyze regional biodiversity conservation and contribute to the development of related industries.

## 2. Materials and Methods

### 2.1. Species Distribution Data

The distribution data for *G. orchidis* were primarily obtained from the Global Biodiversity Information Facility (GBIF, http://www.gbif.org/) and supplemented with field investigations. The species specimens were accessed online through the Chinese Virtual Herbarium (CVH, http://www.cvh.ac.cn). To ensure the timeliness of the specimen data, we selected and recorded plant sample sites from the GBIF within Continental China, using latitude and longitude coordinates. We then cross-referenced the distribution range of the studied species mentioned in the relevant literature [14,15] and supplemented the data with information from the second Qinghai–Tibet scientific survey. To reduce sampling bias, redundant distributed sample points were screened by establishing a 5 × 5 km buffer using ArcGIS 10.8. Within each buffer, one site was randomly retained as a representative point [31,32]. A total of 79 documented records of *G. orchidis* presence in the QTP were compiled and stored in a CSV format (Figure 1).

### 2.2. Environmental Variables

Considering the specific conditions of the research area, this study selected 23 environmental variables from climate, topography, and soil. Climate data, including the last interglacial (LIG, 140,000 yr B.P), Last Glacial Maximum (LGM, 22,000 yr B.P), Mid-Holocene (MH, 6000 yr B.P), current, and future climates, were obtained from the WorldClim Globe Climate Database (http://worldclim.org/) at a resolution of 2.5 arcminutes (ca. 5 km). The time period for the current climate data was 1970–2000, which was used to construct the initial model. Paleo climate data (LIG, LGM, and MH) were generated using the Community Climate System Model version 4 (CCSM4). Future climate data were based on the BCC–CSM2–MR model, which underwent historical tests of CMIP6 and was used in future Shared Socio-economic Pathways (SSPs) scenario tests [33]. Three scenarios under the SSPs were selected as follows: SSP126 (sustainability pathway), SSP370 (medium emission path), and SSP585 (a path dominated by traditional fossil fuels). These scenarios were chosen for their high applicability to the current research and their ability to better evaluate the temperature, precipitation, and atmospheric circulation [34,35]. The future time periods considered were the 2050s (averages for 2041–2060) and 2070s (averages for 2061–2080) [36]. Topographic variables such as slope and aspect were derived using surface analysis tools in ArcGIS. Soil type data were obtained from the Harmonized World Soil Database (HWSD, www.fao.org) and resampled into raster data with a resolution of 2.5 arcminutes (ca. 5 km) to ensure data consistency [37,38]. The biological variables obtained as described above were spatially projected using the World Geodetic System (WGS 1984).

To address potential multicollinearity issues and prevent overfitting of the model predictions, a Pearson correlation analysis was conducted on the environmental variables obtained from WorldClim [39]. The correlation analysis aimed to identify variables that were strongly correlated with each other. Based on the results of the Pearson correlation analysis, the findings were visualized using a correlation heatmap. In the heatmap, darker colors indicate a stronger correlation strength (Figure 2). Then, all the environmental variables were imported into the MaxEnt. Variables with low contribution, as measured by the jackknife in the model, were removed. If two variables had a correlation coefficient (|R|) ≥ 0.8, only the variable with the highest contribution was retained. Ultimately, six variables were selected as the evaluation variables for the model: annual precipitation (Bio12), mean temperature of the coldest quarter (Bio11), minimum temperature of the coldest month (Bio6), temperature seasonality (Bio4), aspect (Asp), and precipitation of the coldest quarter (Bio19) (Table 1). These variables were deemed to have the most significant contribution to the model’s predictions.

### 2.3. Construction, Optimization, and Evaluation of MaxEnt

In this study, the optimized MaxEnt was selected to overcome the complexity and limited portability of the default parameter settings in the initial model [40]. To determine the optimal parameter combination, the ENMeval package in R 4.2.2 was utilized. The regularization multipliers (RMs) were regrouped, ranging from 0.1 to 4.0 with 0.1 intervals, and the feature combination (FC) parameters were considered. The FC parameters included L (Linear), LQ (Quadratic), LQH, H (Fragmentation), LQHP (Product), and LQHPT (Threshold). These parameters were cross-combined to facilitate the evaluation of the MaxEnt’s complexity under different parameter combinations using R 4.2.2 [40,41]. A total of 240 combinations were adjusted using the ENMeval package. Subsequently, the best parameter combination was selected for modeling based on a delta.AICc = 0 [37]. The delta.AICc value was a measure of the difference in the AICc (Akaike Information Criterion corrected for small sample sizes) between models, and a value of 0 indicated the best-fitting model in terms of complexity and goodness of fit. The selected parameter combination was presented in Table 2.

In this study, the MaxEnt was imported with screened occurrence data (* .csv) and environmental variables data (* .asc). The “Logistic” mode was selected as the output mode, which had the same parameter set as the MaxEnt but was more adaptable [42]. To evaluate the predictive power of the models, independent training models were used. The “Bootstrap” method was chosen as the replicated run type, where 25% of the dataset was randomly selected as the test set, and the remaining 75% was used as the training set. This process was repeated 10 times to minimize errors in both the training and test data [43,44]. Sample points were randomly selected from a pool of 10,000 background grid cells, and the “Random seed” option was used to increase the randomness of the model. The plotting data were recorded for generating charts [25,45]. The remaining parameters were kept at their default values. The accuracy of the simulation results was assessed using the receiver operating characteristic (ROC) curve, and the area under the curve (AUC) was calculated. The AUC measures the degree of discrimination between sites where observations were present or absent. A higher AUC value closer to 1 indicates a more accurate model. The forecast results were divided into different intervals based on the AUC values, ranging from excellent (0.9–1.0) to fail (0.5–0.6) [22,46]. After running the model, a suitability index was obtained, representing the range of potentially suitable distribution areas for *G. orchidis*. The suitability index values ranged from 0 to 1, with higher values indicating a higher likelihood of the species being present in that region. Based on the prediction of habitat suitability, the suitability areas were classified into four grades: not suitable (value ≤ MTSPS), generally suitable (MTSPS < value ≤ 0.4), moderately suitable (0.4 < value ≤ 0.6), and highly suitable (0.6 < value) [37]. Furthermore, the SDM toolbox under the ArcGIS extensions was used to analyze the changes in the distribution area and the geometric center of the habitat under different climate change scenarios. This analysis aimed to detect the general trend of the offset in *G. orchidis* distribution.

### 2.4. Application Software and Data Resources

Digital Mountain Map was obtained from the dataset of the Digital Mountain Map of China (2015) [47]; MaxEnt software (Princeton University Press, Princeton, NJ, USA) was downloaded at open source (http://biodiversityinformatics.amnh.org/), versions were 3.4.4. The version of SPSS (IBM, Armonk, NY, USA) was 27.0.

## 3. Results

### 3.1. Model Validation and Importance of Environmental Variables for Prediction

The ROC curves generated from the MaxEnt indicated a high performance in both the training set and the test set. The area under the curve (AUC) value for the training set was 0.939, while the AUC value for the test set was 0.959. These high AUC values, close to 1, suggested that the MaxEnt performed well in the current scenario. The model was able to accurately fit the species distribution data and provide reliable predictions (Figure 3).

The jackknife test results provided insights into the key environmental factors influencing the distribution of *G. orchidis*. The top two variables identified were annual precipitation (Bio12) and mean temperature of coldest quarter (Bio11). These two variables accounted for a cumulative contribution of 78.5%. Additionally, the accrued permutation importance reached 90.1%, further confirming the dominance of Bio12 and Bio11 as significant environmental factors for *G. orchidis*. Furthermore, Bio6 (minimum temperature of coldest month) and Bio4 (temperature seasonality) accounted for up to 15.9% of the contribution, indicating that these variables were equally important for the growth and development of *G. orchidis*. This suggests that factors related to the coldest temperatures and temperature variations throughout the year play a crucial role in determining the suitable habitat for *G. orchidis* (Table 3).

The jackknife test results indicated that the training gain remained stable when each environmental variable was used in isolation, suggesting that these variables individually influenced the predictive ability of the MaxEnt for *G. orchidis*. The variables that affected the suitable area of *G. orchidis* were Bio6, Bio4, Bio19 (precipitation of coldest quarter), Bio12, Bio11, and Aspect. Additionally, the training gain decreased when Bio19 and Aspect were not included, indicating that the predictive power of the model would be reduced if these variables were excluded. This suggested that Bio19 (precipitation of coldest quarter) and Aspect had a profound impact on the suitability of *G. orchidis*. The inclusion of these variables in the model improves its ability to accurately predict the suitable habitat for *G. orchidis* (Figure 4).

### 3.2. Simulated Current Potentially Suitable Distribution

The response curve of *G. orchidis* provided insights into the range of habitat suitability based on the current climatic conditions. According to the response curve, *G. orchidis* showed a preference for specific ranges of environmental variables. For annual precipitation (Bio12), the preferred range was between 613 mm and 2466 mm. This indicated that *G. orchidis* thrives in areas with moderate to high levels of annual precipitation. The mean temperature of the coldest quarter (Bio11) ranged from −5.8 °C to 8.5 °C. *G. orchidis* showed a preference for areas with relatively mild temperatures during the coldest quarter of the year. The temperature seasonality (Bio4) ranged from 437 to 648. *G. orchidis* exhibited a preference for areas with a certain level of temperature variation throughout the year. The minimum temperature of the coldest month (Bio6) ranged from −14.0 °C to 3.7 °C. *G. orchidis* showed a preference for areas where the minimum temperature during the coldest month falls within this range. Lastly, for precipitation of the coldest quarter (Bio19), *G. orchidis* preferred areas with a precipitation level exceeding 11 mm. These ranges provided valuable information for understanding the habitat preferences of *G. orchidis* under current climatic conditions (Figure 5).

Based on the prediction results of the MaxEnt, we used ArcGIS 10.8 techniques to reclassify and visual analytics for the potentially suitable areas of *G. orchidis* in the QTP (Figure 6 and Figure 7). Under the current climatic conditions, the total potential suitable areas of *G. orchidis* reached out to about 63.72 × 10^4^/km^2^, and the proportion of suitable areas that were highly, moderately, generally, and not suitable were 6.55%, 10.33%, 7.33%, and 76.01%, respectively (Table 4).

The potential distribution of *G. orchidis* was found to be primarily located in the southeast region of the QTP, specifically within the latitude range of 25° N to 35° N and longitude range of 80° E to 105° E. This area includes the provinces of Yunnan, Sichuan, Gansu, and parts of Xizang (Tibet). Within this region, the highly suitable areas for *G. orchidis* were predominantly concentrated in the northern part of Yunnan, southern part of Tibet, and eastern regions of Sichuan and Gansu. These highly suitable areas were distributed along mountain ranges such as Hengduan, Yunlin, and the Himalayas. These regions exhibit the most favorable conditions for the growth and development of *G. orchidis*. Moderately and generally suitable areas, which also provided suitable habitat for *G. orchidis*, covered a broader area including the provinces of Yunnan, Gansu, Sichuan, and southern Tibet (Figure 6). These regions may have slightly less optimal conditions compared to the highly suitable areas but still support the growth and survival of *G. orchidis*. Overall, this study identified specific geographic regions within the southeast of the QTP as the potential distribution range for *G. orchidis*, with highly suitable areas concentrated in certain mountainous regions and moderately and generally suitable areas covering a larger extent of the studied provinces.

### 3.3. Predicted Past and Future Potentially Suitable Distribution

The potential suitable area for *G. orchidis* in the past and future remains primarily in the southeast region of the QTP, including the provinces of Yunnan and parts of Tibet, Sichuan, and Gansu. The highly suitable areas continue to be distributed along mountain ranges such as Hengduan, Yunlin, Longmen, and others. During the glacial periods, such as the last interglacial (LIG) and Last Glacial Maximum (LGM), there was less suitability for *G. orchidis* due to the harsh environmental conditions associated with glaciation. However, there was an increase in highly suitable areas during the LGM, primarily focused on mountain ranges. This suggested that *G. orchidis* may have found refuge in these mountainous regions during the glacial periods. In contrast, during the post-glacial period (MH, or Mid-Holocene), the suitable area of *G. orchidis* expanded to the northwest from its previous range. However, the highly suitable areas decreased in this period, and non-mountainous areas gradually became suitable, although more in a generally suitable range. Looking into the future under three scenarios, the suitable area with high suitability for *G. orchidis* was projected to further expand to the northwest of the QTP, specifically in the Qinghai province. However, the southern part of Yunnan is expected to experience a loss of suitability for *G. orchidis* (Figure 7 and Figure 8). These findings suggested that the potential distribution of *G. orchidis* was influenced by past climatic changes and was projected to shift in response to future climate scenarios. The mountainous regions continued to be crucial for the highly suitable habitat of *G. orchidis*, while other areas may become generally suitable or experience changes in suitability.

The suitable area for *G. orchidis* had shown a remarkable growth trend from the past to the future. In the past, the suitable area was increased from 32.56% to 65.79%, having only been 32.56% in the LIG and covered an area of 26.32 × 10^4^/km^2^. However, in the present time, the suitable area has increased to 65.79%, covering a much larger area of 63.72 × 10^4^/km^2^. Looking at the future scenarios, the geographic distribution of *G. orchidis* was projected to expand and contract, but the expansion was expected to outweigh the contraction. Among the three scenarios, SSP370 showed the highest rate of expansion, ranging from 34.07% to 52.2%, while the contraction rate was relatively low, ranging from 3.74% to 6.01%. SSP585 also exhibited expansion, with a range of 14.69% to 38.21%, but there was also a noticeable loss rate and contraction ranging from 1.44% to 22.3%. Overall, there were no significant changes in the overall suitability compared to SSP370. In the SSP126 scenario, there was not much change in the suitability area, but there was less contraction, ranging from 1.77% to 2.21%. In summary, the suitable area for *G. orchidis* showed promising prospects in the future. Among the scenarios, SSP370 provided the best growth environment for *G. orchidis*, followed by SSP585, while SSP126 was the least favorable scenario (Figure 7, Figure 8, Figure 9 and Table 5).

### 3.4. Shift of Distribution Centroid under Different Climate Condition

The geographic coordinate point of the centroid of the suitable region for *G. orchidis* was located in the western part of Basu County (96°42′01″ E, 30°06′13″ N). In the past, the centroid of *G. orchidis* exhibited a remarkable tendency to transfer in a northwest direction, with a migration distance of 244.5/km. Specifically, during the glacial periods, the centroid showed an overall northward migration. However, during the Mid-Holocene (MH), the centroid shifted westward to coordinates 98°05′34″ E, 30°00′14″ N. Under the future climate scenarios, the migration direction of the centroid showed significant differences. In the SSP126 scenario, the centroid moved towards the northeast, indicating a trend of reverse migration in the medium-term, with the farthest migration distance of 104.96/km compared to other scenarios. In the SSP370 scenario, the centroid shifted westward to Bomi County (96°15′34″ E, 30°02′39″ N), with the least offset distance of 42.92/km. In the SSP585 scenario, the center point initially moved towards the north but eventually shifted northeastward to Chaya County (97°36′09″ E, 30°34′05″ N), which was close to the centroids in the SSP126 scenario. These migration patterns of the centroid point indicate the shifting distribution of the suitable habitat for *G. orchidis* under different future climate scenarios (Table 6, Figure 10).

## 4. Discussion

Hydrothermal conditions have been identified as the most important variables influencing the geographical distribution of plants, particularly terrestrial plants [35]. These conditions also play a key role in explaining the association between species richness and terrain [48]. In our study, we found that annual precipitation (Bio12) and mean temperature of coldest quarter (Bio11) were the dominant variables influencing the distribution of *G. orchidis*. This suggests that temperature and rainfall are the driving mechanisms for *G. orchidis*. Secondary variables such as min temperature of coldest month (Bio6), temperature seasonality (Bio4), and precipitation of coldest quarter (Bio19) further indicate that rainfall and temperature changes during the coldest quarter can influence the growth of *G. orchidis*. These findings align with the overall habitat characteristics of the *Orchidaceae* family [49] (Table 1). This study found that *G. orchidis* has a limit threshold for cold and drought resistance at −14.0 °C and 11 mm, respectively; this phenomenon is likely a result of the combined effects of the uplift of the plateau during the Neogene period (Miocene and Pliocene) and the climatic fluctuations throughout the Quaternary period. These geological and climatic factors have played a crucial role in shaping the evolution of alpine plants, enabling them to develop unique traits such as cold and drought resistance [16]. Furthermore, it also found that *G. orchidis* shows a preference for southern slopes and an annual precipitation of 900 mm. This phenomenon highlights that the suitable habitat for these plants tends to be in regions with moderate sunlight exposure and ample availability of water and favorable thermal conditions [50]. This could be attributed to the fact that sunlight exposure can increase the production of chemical compounds in *G. orchidis* and promote its growth and development [9]. Additionally, studies have suggested that the mountains on the southern slope of the study area experience higher temperatures and precipitation compared to the northern slope [51,52,53]. This indicates that the southern slope provides a sheltered environment to resist extreme cold conditions, which may be related to the diverse climate types in this region [49]. It is important to note that this study focused solely on the influence of climate, topography, and soil type on species distribution and did not consider biological factors such as interspecific relationships and anthropogenic forcing. Other factors, including genetic variation and soil environments, should be considered in future studies [35,49,54,55,56]. Therefore, further research is needed to explore the dominant factors affecting *G. orchidis*.

The moderate and high mountainous areas in regions with high precipitation, temperature, and humidity provide suitable habitats for rare species, including *G. orchidis* [57]. Our study predicts that the potential suitable areas for *G. orchidis* will change under different climate scenarios over time, showing significant fluctuations in suitability. The results from the MaxEnt indicate that the distribution range of *G. orchidis* along mountains such as Henduan, Yunlin, and Longmen has consistently expanded, especially during the Mid-Holocene (MH) when increased precipitation driven by solar radiation occurred [15]. Previous studies have also suggested that terrestrial herbs gradually became dominant on the QTP during the MH. The northwestward movement of the species’ centroids further indicates the expansion of suitable areas for *G. orchidis* (Figure 10) [13]. During the glacial period, the suitability for *G. orchidis* was reduced, likely due to extensive ice coverage on the QTP [58]. The increase in highly suitable areas of *G. orchidis* during the LGM was more likely attributed to the southern slope of the mountains serving as a glacial refuge for *G. orchidis* [51]. Under different future scenarios, the suitable areas for *G. orchidis* continue to expand towards the northwest, including the provinces of Qinghai and Tibet, particularly in the SSP370 scenario (projecting a temperature rise of 3.6 °C by the year 2100), where the expansion ratio reaches 46.19%. However, there is a loss of suitable areas in the SSP126 scenario (projecting a temperature rise of 1.8 °C by the year 2100), with contraction areas mainly concentrated in the Himalayas, possibly due to reduced precipitation [50,59]. Significant changes are observed in the SSP585 scenario (projecting a temperature rise of 4.4 °C by the year 2100), with the rate of change in suitable areas ranging from 1.4% to 22.3%. The centroid of *G. orchidis* in the 2050s approaches the centroid of the SSP126 scenario in the 2070s, indicating that the climatic conditions of the SSP126 scenario in the 2070s may be similar to those in the 2050s under the SSP585 scenario on the QTP. The multidimensional migration of centroids also suggests that changes in suitability may be influenced by carbon dioxide (CO_2_) emissions [33,36,60]. Increased CO_2_ concentrations can improve the survival of *G. orchidis* and align with the future growth trend of the Orchidaceae family as a whole [49]. However, human activities can disrupt suitable areas and force species migration, which is a complex issue. If a suitable environment is not found during the migration process, *G. orchidis* may be at risk of extinction [57,61]. As an extremely rare and environmentally sensitive Tibetan medicinal resource, the habitat conditions of *G. orchidis* are scarce and difficult to recover once lost. Therefore, proactive measures should be taken to reduce emissions and protect existing habitats.

ENMeval, as a criterion for quantitative evaluations of ecological niche models (ENMs), is widely used in optimizing the MaxEnt to avoid multicollinearity and improve the accuracy of predictions. In our study, we utilized an optimized MaxEnt to simulate and predict the suitability of *G. orchidis* under different scenarios. The high values of the AUC (area under the curve) greater than 0.9 indicate an excellent predictive performance. However, it is important to note that MaxEnts are based solely on existing data, which may span a large period of time. They may lack consideration for actual conditions such as biological interactions, genetic variability, and species dispersal capacity, which can result in a wider prediction range for species distribution areas [55]. Therefore, future studies could further refine the model and incorporate more field validation to obtain more comprehensive details about the species, leading to better conservation assessments.

## 5. Conclusions

In this study, the optimized MaxEnt was used to analyze the changes in the distribution pattern of *G. orchidis* in ancient and modern periods and make predictions for suitable areas under multiple future scenarios. The results highlighted that annual precipitation (Bio12) and mean temperature of the coldest quarter (Bio11) were the dominant factors, accounting for 55.1% and 23.4% of the influence, respectively. The range of Bio12 varied from 613 mm to 2466 mm, while Bio11 ranged from −5.8 °C to 8.5 °C. This study revealed a significant expansion of the suitable areas for *G. orchidis* from the past to the future. In the past, mountains such as Henduan, Yunlin, and Longmen served as glacial shelters for *G. orchidis*, and the centroids of its distribution shifted northward. Furthermore, the trend of northward migration is projected to persist. Under the future scenarios, there was a substantial expansion in the SSP370 scenario (30.33% to 46.19%), followed by the SSP585 scenario (1.41% to 22.3%). However, there was a contraction in suitable areas under the SSP126 scenario. The centroids of *G. orchidis* exhibited multidirectional movement, with the greatest distance observed in the SSP585 scenario (100.38/km). In summary, *G. orchidis* is expected to have better prospects in the future, but it is crucial to preserve its suitable habitats and avoid human-induced destruction. The conservation of *G. orchidi*s not only has profound impacts on economic development but also plays a crucial role in safeguarding biodiversity and maintaining the stability of ecosystem structures worldwide.

## Figures and Tables

**Figure 1 plants-13-00645-f001:**
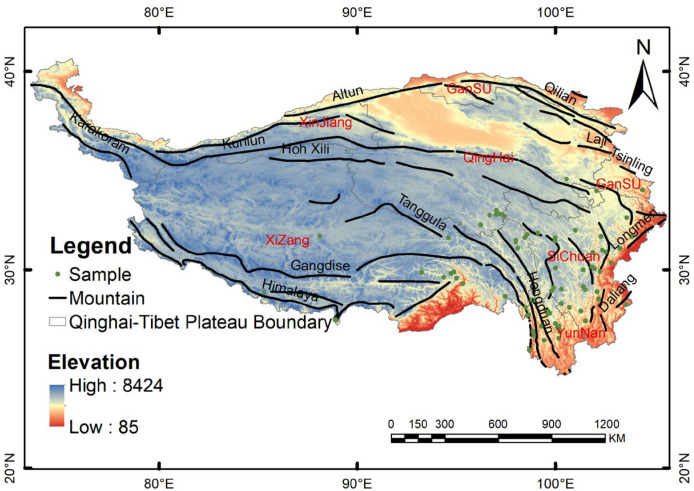
Qinghai–Tibet Plateau distribution of *G. orchidis* sample points.

**Figure 2 plants-13-00645-f002:**
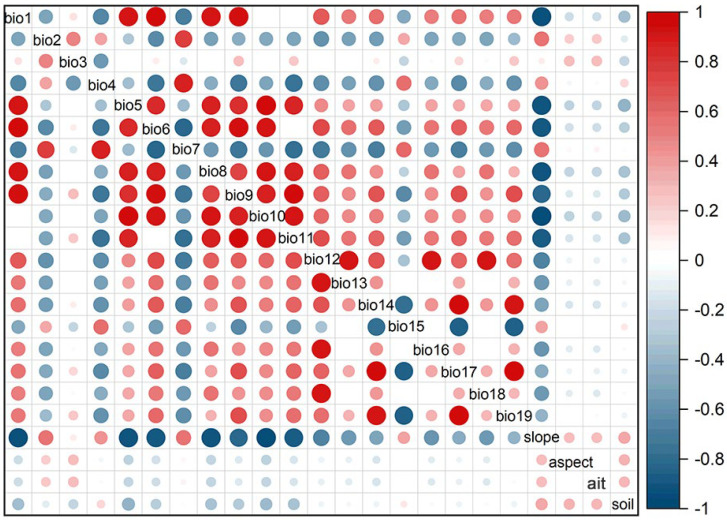
Correlation diagram of environmental variables for the *G*. *orchidis*.

**Figure 3 plants-13-00645-f003:**
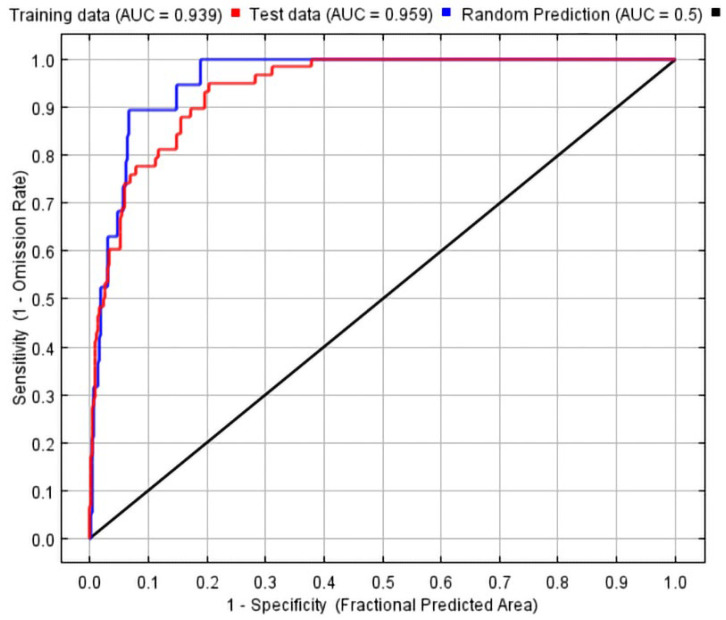
Receiver operating characteristic curve with the area under the curve (AUC).

**Figure 4 plants-13-00645-f004:**
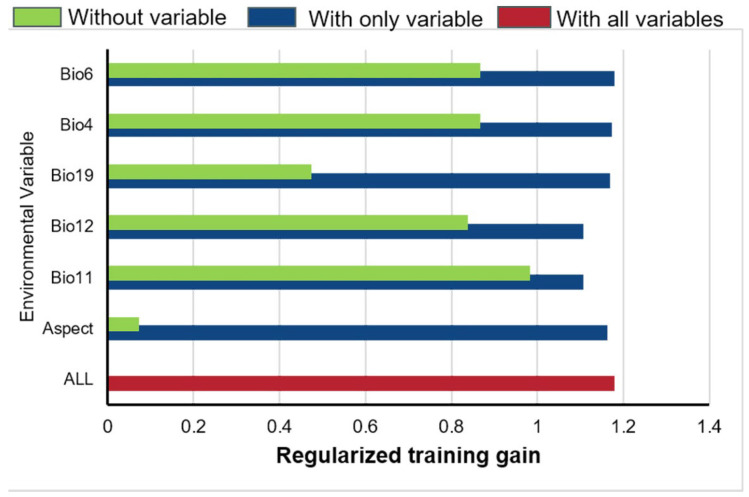
Jackknife plot for training gain.

**Figure 5 plants-13-00645-f005:**
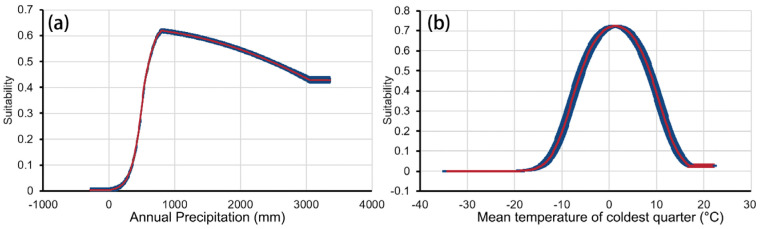
Response curves of *G. orchidis*. (**a**) Annual precipitation and (**b**) mean temperature of coldest quarter.

**Figure 6 plants-13-00645-f006:**
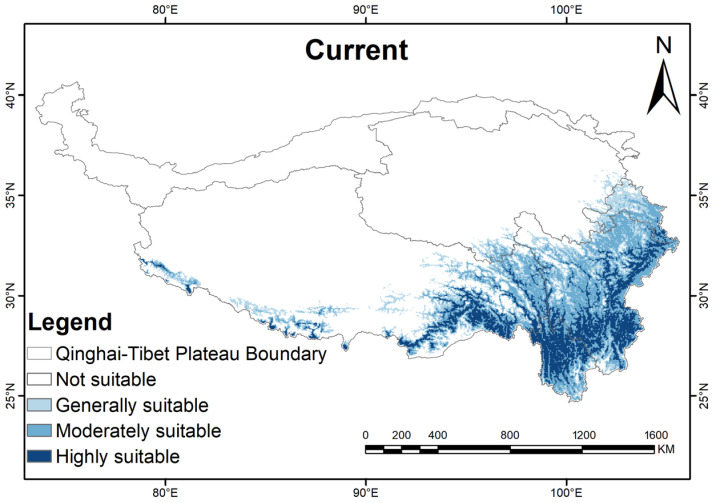
Distribution of potential suitable areas of the QTP region of *G.orchidis* under the current climate scenario.

**Figure 7 plants-13-00645-f007:**
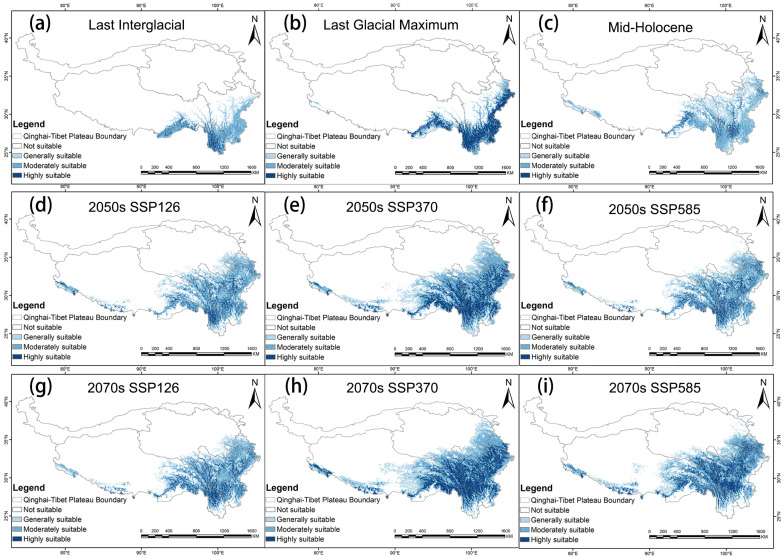
Distribution of potential suitable areas of the QTP region of *G. orchidis* under different climate change scenarios in the future. (**a**) The potential geographical distribution of *G. orchidis* under the last interglacial. (**b**) The potential geographical distribution of *G. orchidis* under the Last Glacial Maximum. (**c**) The potential geographical distribution of *G. orchidis* under the Mid-Holocene. (**d**) The potential geographical distribution of *G. orchidis* under the SSP126 in the 2050s. (**e**) The potential geographical distribution of *G. orchidis* under the SSP370 in the 2050s. (**f**) The potential geographical distribution of *G. orchidis* under the SSP585 in the 2050s. (**g**) The potential geographical distribution of *G. orchidis* under the SSP126 in the 2070s. (**h**) The potential geographical distribution of *G. orchidis* under the SSP370 in the 2070s. (**i**) The potential geographical distribution of *G. orchidis* under the SSP370 in the 2070s.

**Figure 8 plants-13-00645-f008:**
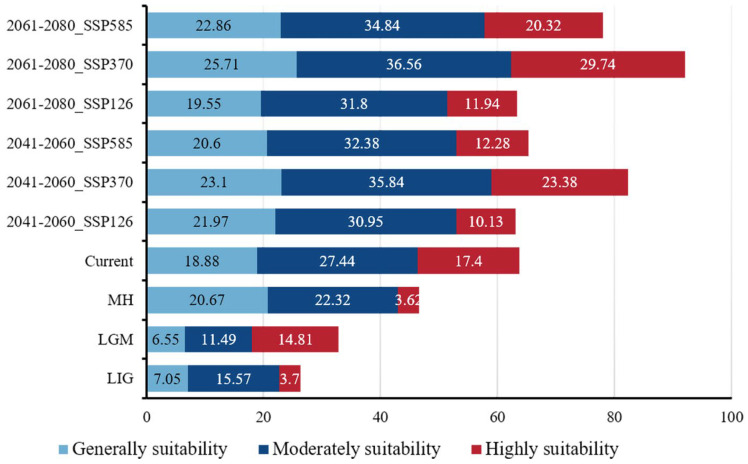
Potential suitable area of *G. orchidis* under different scenarios (×10^4^/km^2^).

**Figure 9 plants-13-00645-f009:**
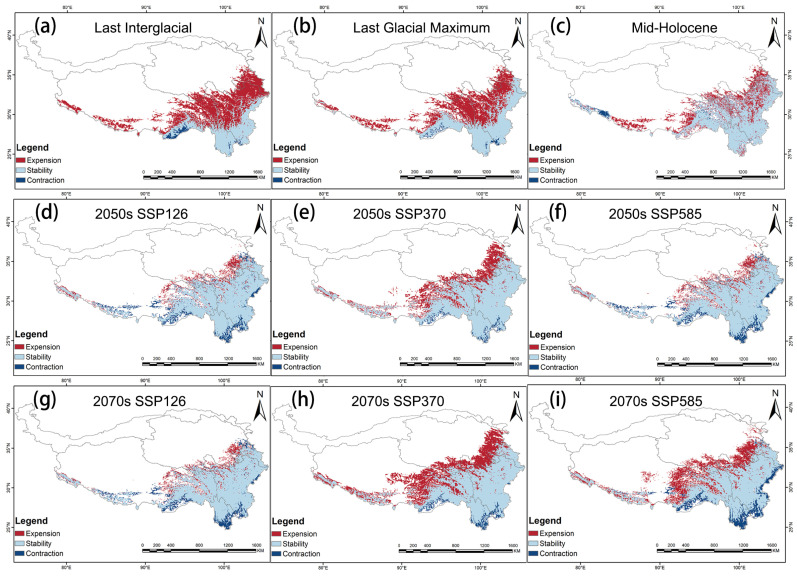
Spatial variation in potential habitat of *G. orchidis* under different climate scenarios compared with modern climate scenarios. (**a**) Area changed under the last interglacial, (**b**) area changed under the Last Glacial Maximum, (**c**) area changed under the Mid-Holocene, (**d**) area changed under the SSP126 in the 2050s, (**e**) area changed under the SSP370 in the 2050s, (**f**) area changed under the SSP585 in the 2050s, (**g**) area changed under the SSP126 in the 2070s, (**h**) area changed under the SSP370 in the 2070s, (**i**) area changed under the SSP585 in the 2070s.

**Figure 10 plants-13-00645-f010:**
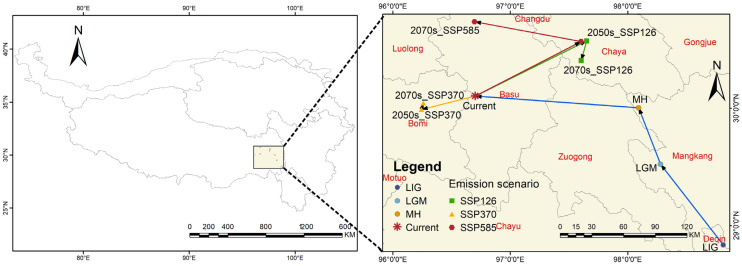
Centroid transfer in potential suitable areas of *G. orchidis* under different climate scenarios.

**Table 1 plants-13-00645-t001:** Environment variables for driving MaxEnt.

Code	Variables	Unit
Bio12	Annual precipitation	mm
Bio11	Mean temperature of coldest quarter	°C
Bio6	Min temperature of coldest month	°C
Bio4	Temperature seasonality (standard deviation × 100)	Dimensionless
Aspect	Aspect based on a digital elevation model	degree
Bio19	Precipitation of coldest quarter	mm

**Table 2 plants-13-00645-t002:** MaxEnt optimized parameter setting by ENMeval.

Type	FC	RM	Delta.AICc	Mean.OR10
Default	LQHP	1	63.07	0.25
Optimized	LQH	2.6	0	0.21

**Table 3 plants-13-00645-t003:** Contribution rate of environmental variables of *G. orchidis*.

Variables	Percent Contribution (%)	Permutation Importance (%)
Bio12	55.1	16.8
Bio11	23.4	73.3
Bio6	8.1	0.1
Bio4	7.8	2.5
Aspect	3.8	4.1
Bio19	1.9	3.2

**Table 4 plants-13-00645-t004:** Change in current potential suitable area of *G. orchidis*.

Year	Suitable Area/×10^4^/km^2^			
	Not Suitable	Generally Suitable	Moderately Suitable	Highly Suitable
1970–2000	201.92	18.88	27.44	17.4

**Table 5 plants-13-00645-t005:** Suitable distribution of *G. orchidis* under different climate change scenarios.

Year	Area/×10^4^ km^2^				Change Rate/%			
	Increase	Reserved	Last	Change	Increase	Reserved	Last	Change
LIG	41.92	26.12	1.97	39.95	65.79	40.99	3.09	62.70
LGM	33.92	34.09	1.28	32.64	53.23	53.50	2.01	51.22
MH	20.75	47.29	2.14	18.61	32.56	74.22	3.36	29.20
2050s_SSP126	7.93	58.70	9.34	−1.41	12.45	92.12	14.66	−2.21
2050s_SSP370	21.71	63.00	2.38	19.33	34.07	98.87	3.74	30.33
2050s_SSP585	9.36	59.58	8.46	0.90	14.69	93.50	13.28	1.41
2070s_SSP126	7.59	59.32	8.72	−1.13	11.91	93.09	13.68	−1.77
2070s_SSP370	33.26	63.72	3.83	29.43	52.20	100.00	6.01	46.19
2070s_SSP585	24.35	57.91	10.14	14.21	38.21	90.88	15.91	22.30

**Table 6 plants-13-00645-t006:** Centroid coordinates and migration distance.

Climate Scenario	Year	Longitude	Latitude	Migration Distance/km
	LIG	98°48′49″ E	28°50′03″ N	244.55
	LGM	98°16′44″ E	29°31′19″ N	165.47
	MH	98°05′34″ E	30°00′14″ N	134.48
	Current	96°42′01″ E	30°06′13″ N	0.00
SSP126	2050s	97°38′56″ E	30°34′24″ N	104.96
	2070s	97°36′12″ E	30°24′25″ N	93.07
SSP370	2050s	96°14′32″ E	29°59′24″ N	45.85
	2070s	96°15′34″ E	30°02′39″ N	42.92
SSP585	2050s	97°36′09″ E	30°34′05″ N	100.83
	2070s	96°41′42″ N	30°44′13″ N	70.42

## Data Availability

Data are contained within the article.

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
