# Peer review of "Prediction of Historical, Current, and Future Configuration of Tibetan Medicinal Herb Gymnadenia orchidis Based on the Optimized MaxEnt in the Qinghai–Tibet Plateau"

_plants, 2024, doi:10.3390/plants13050645_

Round 1

Reviewer 1 Report

Comments and Suggestions for Authors

General Comments

The authors investigated the "Prediction of historical, current, and future configuration of Tibetan medicinal herbs Gymnadenia orchidis based on the optimized MaxEnt in the Qinghai-Tibet Plateau." However, the authors are advised to add more relevant literature to introduce the effects of spatial configuration on medicinal herbs at high plateaus.

Besides, it is strongly suggested to explain how different environmental conditions affect the growth and quality of medicinal plants, especially the medicinal contents. If possible, for authors, then please compare the growth and accumulation of medicinal contents in medicinal herbs under different growing conditions, e.g., low-, medium-, and high-rainfall conditions and photosynthetically active radiation zones.

Then, the message of this manuscript will be more effective and clearer for readers. As a whole, the introduction and discussion sections lack reliable ecological and physio-chemical explanations to explain how different environmental conditions affect herb growth in high-elevation areas or regions. Therefore, I would recommend a minor revision of the manuscript.

Specific Comments

Abstract

The abstract needs to be revised (e.g., provide research background based on your research question and then link your research question and obtained results systematically with the aims and objectives of this study. Moreover, the conclusion of the abstract also needs some modifications. The conclusion generally provides the future research directions. 

Introduction

The introduction does not provide enough information on the research background and gap. Therefore, the authors advised adding more relevant information because an introduction always needs to be very clear on what we already know, what we don't know, and which questions are, therefore, addressed by the current research.

Discussion

I think authors should rethink what they write in the first paragraph and only summarize the main findings in view of the research questions. After this, authors can explore different aspects of the work in subsequent paragraphs and explain how their findings expand the envelope of knowledge, but first of all, authors need to state the main results without discussing their why and how or the relationships to the literature. First of all, the reader needs a clear statement of what the study found. 

Reviewer 2 Report

Comments and Suggestions for Authors

Dear author, your manuscript, "Prediction of historical, current, and future configuration of Tibetan medicinal herbs Gymnadenia orchidis based on the optimized MaxEnt in the Qinghai-Tibet Plateau," presents a valuable exploration. The following suggestions aim to enhance clarity, coherence, and specificity throughout the text.

Abstract:

1. Line 13 and line 42: Please include the author's name for Gymnosia orchidis.

2. Line 38: Avoid duplicating words from the title in the list of keywords.

3. Lines 53-58: Edit the sentence to read, "The temporal and spatial heterogeneity of G. orchidis is pronounced, with suitability continuously expanding from MH (Mid-Holocene). The major distribution areas are situated in Qinghai, Yunnan…"

4. Lines 63-64: Consider rephrasing for clarity. I suggest, "Moreover, the Qinghai-Tibet Plateau (QTP) functions as a highly sensitive and vulnerable ecological screen to climate variability, not only in China but also across Asia as a whole."

5. Lines 84-87: Rephrase for clarity. Also, provide more specific information when mentioning the Qinghai-Tibet Plateau regarding its significance to G. orchidis distribution.

6. In the introduction, use transition phrases to connect ideas smoothly between sentences and paragraphs.

7. The manuscript focuses on predicting the distribution of medicinal species, specifically Gymnosia orchidis, using SDM. Enhance the manuscript by incorporating more relevant citations that highlight the application of SDM in predicting the distribution of medicinal plant species. I recommend the following citation:

   - Rahmanian, S., Pouyan, S., Karami, S., & Pourghasemi, H. R. (2022). Predictive habitat suitability models for Teucrium polium L. using boosted regression trees. In Computers in Earth and Environmental Sciences (pp. 245-254). Elsevier.

   - Yang, L., Zhu, X., Song, W., Shi, X., & Hang, X. (2023). Predicting the potential distribution of 12 threatened medicinal plants on the Qinghai-Tibet Plateau (QTP) with a maximum entropy model.

Method:

1. Provide the text that cites Fig. 2, and then include Fig. 2 in the document.

2. Lines 139-143: Begin these words with lowercase letters: annual precipitation, mean temperature of the coldest quarter, minimum temperature, temperature, aspect, precipitation...

3. Clarify the explanation of how variables were selected (lines 134-141).

4. Establish a smoother transition between the results section and the discussion of past and potentially future suitable distributions. Consider including a sentence summarizing the main findings to aid in this transition.

Comments on the Quality of English Language

The clarity and fluency of the English language in the manuscript need improvement. Consider revising for better readability and coherence.
